# Cardioprotective Role for Paraoxonase-1 in Chronic Kidney Disease

**DOI:** 10.3390/biomedicines10092301

**Published:** 2022-09-16

**Authors:** Prabhatchandra Dube, Fatimah K. Khalaf, Armelle DeRiso, Chrysan J. Mohammed, Jacob A. Connolly, Dhanushya Battepati, Apurva Lad, Joshua D. Breidenbach, Andrew L. Kleinhenz, Bella Khatib-Shahidi, Mitra Patel, Iman Tassavvor, Amira F. Gohara, Deepak Malhotra, Eric E. Morgan, Steven T. Haller, David J. Kennedy

**Affiliations:** 1Department of Medicine, College of Medicine and Life Sciences, University of Toledo, Toledo, OH 43606, USA; 2Department of Clinical Pharmacy, University of Alkafeel, Najaf 54001, Iraq; 3Department of Surgery, University of Toledo College of Medicine and Life Sciences, Toledo, OH 43606, USA

**Keywords:** paraoxonase-1, cardiac hypertrophy, left ventricular function, inflammation, fibrosis

## Abstract

Paraoxonase-1 (PON-1) is a hydrolytic enzyme associated with HDL, contributing to its anti-inflammatory, antioxidant, and anti-atherogenic properties. Deficiencies in PON-1 activity result in oxidative stress and detrimental clinical outcomes in the context of chronic kidney disease (CKD). However, it is unclear if a decrease in PON-1 activity is mechanistically linked to adverse cardiovascular events in CKD. We investigated the hypothesis that PON-1 is cardioprotective in a Dahl salt-sensitive model of hypertensive renal disease. Experiments were performed on control Dahl salt-sensitive rats (SS^Mcwi^, hereafter designated SS-WT rats) and mutant PON-1 rats (SS-Pon1^em1Mcwi^, hereafter designated SS-PON-1 KO rats) generated using CRISPR gene editing technology. Age-matched 10-week-old SS and SS-PON-1 KO male rats were maintained on high-salt diets (8% NaCl) for five weeks to induce hypertensive renal disease. Echocardiography showed that SS-PON-1 KO rats but not SS-WT rats developed compensated left ventricular hypertrophy after only 4 weeks on the high-salt diet. RT-PCR analysis demonstrated a significant increase in the expression of genes linked to cardiac hypertrophy, inflammation, and fibrosis, as well as a significant decrease in genes essential to left ventricular function in SS-PON-1 KO rats compared to SS-WT rats. A histological examination also revealed a significant increase in cardiac fibrosis and immune cell infiltration in SS-PON-1 KO rats, consistent with their cardiac hypertrophy phenotype. Our data suggest that a loss of PON-1 in the salt-sensitive hypertensive model of CKD leads to increased cardiac inflammation and fibrosis as well as a molecular and functional cardiac phenotype consistent with compensated left ventricular hypertrophy.

## 1. Introduction

Cardiovascular morbidity and mortality are very common among patients with chronic kidney disease (CKD) [1,2,3,4]. As CKD progresses, renal disease-specific risk factors for cardiovascular events become operant. As a result, patients with CKD often develop severe abnormalities in their cardiac physiological function as well as significant cardiac hypertrophy and fibrosis [1,2]. By postulating that kidney disease is the main disorder and cardiovascular complications are tributary, Bright described the concept of the renal origin of cardiovascular disease [5]. Numerous studies have examined and reported this association [4,6]. We and others have demonstrated a clinical association between decreases in circulating paraoxonase-1 (PON-1) enzyme activity and poor cardiovascular outcomes in CKD settings [7,8]. In the current study, we sought to examine if PON-1 plays a mechanistic role in the pathophysiology of cardiovascular disease associated with CKD.

PON-1 is a calcium-dependent hydrolytic enzyme that is produced in the liver and bound to high-density lipoprotein (HDL) cholesterol [9]. Many studies have shown the important role of PON-1 in cardiovascular disease (CVD) [10]. Studies examining transgenic PON-1 knockout models demonstrated the capability of PON-1 to protect against atherosclerotic events through inhibiting atherogenesis [11]. HDL’s effect in lowering low-density lipoprotein (LDL) lipid peroxidation, which is mainly mediated through PON-1 activity, was found to persist longer than the effect of antioxidant vitamins, and hence, was deemed more protective [12,13]. A number of other significant clinical studies in the field of PON research have occurred. Observational and case–control studies have demonstrated an association between coronary heart disease (CHD) and decreased serum PON-1 activity [14]. This correlation has been additionally confirmed by prospective studies demonstrating that decreased serum PON-1 activity is an independent risk factor that could predict future CHD events [15]. PON-1 has also been shown to be most active and effective in lipid peroxide hydrolysis [16,17,18,19]. In addition, evidence proposes a possible role for PON-1 in potentiating CVD associated with chronic kidney disease. Many studies have demonstrated an association between low-circulating PON-1 activity and adverse CVD outcomes in CKD settings [7,8,9,10]. In fact, lower PON-1 levels were associated with higher CRP, lower HDL3, and adverse future CVD outcomes [11,12]. Altered PON-1 activities in different HDL subclasses could be essential factors in the development of atherosclerosis in patients with CKD as well as end-stage kidney disease [20]. Patients with CKD have decreased circulating levels of both the PON-1 protein and PON lactonase activity, as well as increased cardiovascular morbidity and mortality compared to control individuals [8]. Despite these clinical findings, the exact pathophysiologic mechanism whereby low-circulating PON-1 leads to poor CVD outcomes in CKD is not fully understood. Further, whether decreased PON-1 has direct effects on cardiac function or cardiac injury in CKD, in addition to its well described role in atherogenesis, is unknown. Based on this background, we aimed to examine the role of PON-1 as a possible factor in mediating cardiac disease in a well-established model of high-salt-mediated CKD. We tested the hypothesis that decreased PON-1 is mechanistically linked to cardiac injury and dysfunction in CKD.

## 2. Materials and Methods

### 2.1. Animals

All animal studies were performed in accordance with the National Institutes of Health’s Guide for the Care and Use for Laboratory Animals and approved by the Institutional Animal Care and Use Committee at the University of Toledo. Control Dahl salt-sensitive rats (SS^Mcwi^, henceforth called SS-WT rats) as well as PON-1 mutant rats (SS-PON-1em1Mcwi, henceforth called SS-PON-1 KO rats) were generated by the Medical College of Wisconsin Gene Editing Rat Resource Center as we have previously described [21]. Briefly, SS-PON-1 KO rats were created via CRISPR insertion by directing the sequence AGTATTTTTCCAGGCTTACTGG into embryos of SS rats. The subsequent mutation was a 7bp frameshift insertion in Exon 4. Animals were genotyped using a Cel-1 assay followed by Sanger sequencing to confirm the founder animals. Animals were further reciprocally backcrossed to the paternal strain and successive litters’ genotype were confirmed via fluorescent genotyping. Recipient animals were Sprague Dawley rats from Charles River. Ten-week-old male rats were maintained at the University of Toledo Department of Animal Laboratory Resources for five weeks on either a normal chow (NC) diet or high-salt (HS) diet (8% NaCl, Envigo, Teklad diets, Madison, WI, USA) to induce salt-sensitive hypertensive renal disease that is distinctive of this model. The genotype of all animals enrolled in the study protocol was confirmed by DNA sequencing.

### 2.2. Echocardiography

The left ventricular geometry and global systolic function of SS-PON-1 KO and SS-WT rats were evaluated via echocardiography utilizing previously well described methods [22]. Briefly, 2D-guided M-mode images of the left ventricle were obtained in the parasternal long-axis window, and Doppler interrogation of the aortic inflow was performed from a foreshortened apical window. The maximal LVOT diameter was determined in parasternal long-axis view. The LV geometry was assessed by measuring end-diastolic and -systolic dimensions and LV and septal wall thickness at the mid-chamber, and by calculating relative wall thickness. The global systolic function was evaluated by determining fractional shortening, the velocity of circumferential shortening, and the cardiac index as previously described [22].

### 2.3. Histology

Hearts were fixed in 4% formaldehyde (pH 7.2), paraffin embedded, and cut into 4 μm sections. Following this process, tissue slices were deparaffinized with xylene and then rehydrated by consecutive incubations in ethanol and water. CD68 antibodies were purchased from Abcam (Cambridge, MA, USA). A Vectastain Elite-ABC Kit was purchased from Vector Labs (Burlingame, CA, USA). H&E and trichrome staining for the 4 μm heart sections were then performed. Ten images were randomly taken for each section, using a bright-field microscope with a 20× lens. Quantitative analysis was then conducted using an automated and customized algorithm/script for batch analysis (ImageIQ Inc., Cleveland, OH, USA), as described in detail in [8]. The heart histology was then blindly graded by a pathologist (A.G.) and scored using a semiquantitative scale of 0 to 4.

### 2.4. Reverse Transcription-Polymerase Chain Reaction (RT-PCR) and RNA Isolation

Ribonucleic acid (RNA) extraction, complementary DNA (cDNA) synthesis, and RT-PCR were all accomplished using the automated workflow system from QIAGEN (Germantown, MD, USA). RNA was isolated from heart tissue using the QIAzol/chloroform extraction methodology via automated equipment (QIAcube HT). Roughly 500 ng of isolated RNA was used to prepare cDNA (QIAGEN’s RT2 First Strand Kit, cat. #330404), which was followed by using QIAGEN’s Rotor-Gene Q thermocycler to perform RT-PCR. Gene expression was calculated by comparing the relative change in the cycle threshold value (ΔCt). The 2^−ΔΔCt^ equation was then used to calculate the fold change in the gene expression as previously described [21].

### 2.5. Statistical Analysis

GraphPad Prism 6 software was used to perform all statistical analyses of the current study. Data are presented as the mean ± standard error of the mean. Student’s unpaired *t*-test was used to evaluate statistically significant differences between two groups, whereas a one-way ANOVA and post hoc tests were used for comparing more than two groups. Statistical significance was considered when *p* < 0.05.

## 3. Results

### 3.1. PON-1 KO Rats on High-Salt Diet Demonstrate LV Cardiac Hypertrophy

The echocardiographic evaluation at baseline (on a normal chow diet) showed no differences in left ventricular geometry (as assessed by end-diastolic and -systolic dimensions, LV and septal wall thickness, and relative wall thickness) between the SS-PON-1 KO NC and age-matched SS-WT NC. Similarly, there were no differences in global systolic function (as measured by fractional shortening, the velocity of circumferential shortening, and the cardiac index) (Figure 1). After 4 weeks on an 8% high-salt diet, however, SS-PON-KO HS rats demonstrated an increase in LV and septal wall thickness and a significantly decrease in end-diastolic and -systolic dimensions, yielding a significantly increased relative wall thickness, which is consistent with the development of concentric LV hypertrophy (Figure 2). These changes were accompanied by an increase in global systolic function as evidenced by increases in fractional shortening, the velocity of fractional shortening, and the cardiac index. (Figure 2). 

We next examined excised hearts from SS-PON-1 KO and SS-WT rats to directly assess for cardiac hypertrophy. We noted that the heart-weight-to-body-weight ratio was significantly higher in SS-PON-1 KO HS rats compared to the age-matched SS-WT HS rats (Figure 3A). In addition, we performed the RT-PCR on hearts from both SS-PON-1 KO and SS-WT rats to examine the expression levels of genes that are associated with cardiac hypertrophy. RT-PCR analysis of left ventricular tissue revealed a significant increase in the expression of the natriuretic peptide A and B-myosin heavy chain in SS-PON-1 KO HS rats compared to SS-WT HS rats after 5 weeks of an 8% high-salt diet (Figure 3B).

### 3.2. SS-PON-1 KO Rats on High-Salt Diet Demonstrate Significantly Increase in Cardiac Fibrosis Compared to Age-Matched SS-WT Rats

Heart sections were stained to inspect histological evidence of cardiac injury. Hearts of SS-PON-1 KO HS rats showed a significant increase in cardiac fibrosis compared to SS-WT HS rats as assessed by trichrome staining (Figure 4A). The RT-PCR array performed on hearts from SS-PON-1 KO HS rats showed a significant upregulation of the tissue inhibitor of metalloprotease-1 expression compared to SS-WT HS rats, which is associated with cardiac fibrosis and remodeling (Figure 4B).

### 3.3. SS-PON-1 KO Rats on High-Salt Diet Demonstrate Significantly Increase in Cardiac Inflammation Compared to SS-WT Rats

In order to evaluate cardiac inflammation, heart sections from SS-PON-1 KO and SS-WT rats were stained for CD68-positive (i.e., macrophage) immune cells. Upon analysis, we noted a significant increase in macrophage infiltration in both perivascular and interstitial regions within the heart sections of SS-PON-1 KO HS rats compared to SS-WT HS rats after 5 weeks of an 8% high-salt diet (Figure 5A). Following this, we conducted the RT-PCR on hearts from SS-PON-1 KO and SS-WT rats to examine inflammatory gene expression. SS-PON-1 KO HS compared to SS-WT HS rats demonstrated significant upregulation of CCL2/MCP1 gene expression, which is indicative of increased cardiac inflammation and immune cell recruitment (Figure 5B). These results are consistent with our histological analysis with H&E staining of cardiac sections, which demonstrated increases in interstitial immune cell infiltration and cardiac inflammation in SS-PON-1 KO HS compared to SS-WT HS rats after 5 weeks of an 8% high-salt diet (Figure 5C).

### 3.4. SS-PON-1 KO Rats on High-Salt Diet Demonstrate Significantly Decreased Calcium Handling Genes Compared to SS-WT Rats

Finally, we performed a RT-PCR on hearts from SS-PON-1 KO and SS-WT rats to examine the expression of key calcium cycling genes [23]. We noted that SS-PON-1 KO HS rats showed a significant decrease in sarcoplasmic reticulum calcium ATPase 2A (Atp2a2 or SERCA2a) and sodium–calcium exchanger (SLC8A1 or NCX-1) gene expression as compared to SS-WT HS rats after 5 weeks of an 8% high-salt diet (Figure 6).

## 4. Discussion

Despite advances in CKD management strategies, care for these patients is often complicated by their susceptibility to developing CVD [24]. CVD is the foremost cause of death in CKD patients [25]. Although current treatment strategies to prevent cardiac disease in CKD patients have been focused on the neurohormonal blockade, augmenting endogenous counter-regulatory defense mechanisms may be a more successful strategy. Along these lines, there are multiple biomarkers associated with the progression of CVD in CKD patients that have been suggested as potential therapeutic targets [26]. The dysfunctional high-density lipoprotein (HDL) has emerged as a potential central theme in contributing to the disproportionate cardiovascular morbidity and mortality in CKD [27]. PON-1 is an enzyme that is produced in the liver and circulates by being bound to high-density lipoprotein (HDL) cholesterol [28]. The anti-atherogenic and antioxidant features of the HDL are well recognized and due, in part, to PON-1 [29,30]. In fact, several lines of clinical evidence suggest that diminished circulating PON-1 is in fact associated with adverse cardiovascular outcomes in CKD [7,8].

PON-1 has the capacity to hydrolyze oxidized the LDL and cleave phospholipid peroxidation adducts [31]. Ex vivo studies demonstrate that PON-1 is very critical in enhancing the capacity of the HDL to protect the LDL from oxidative modification; this could explain some of the relationship between PON-1 and CVD in clinical settings [32]. Serum PON-1 activity and concentration are decreased in patients with CHD compared to controls [31]. Reduced PON-1 activity has also been associated with an increased risk for CVD and adverse outcomes [33].

With respect to animal studies, human PON-1 transgenic mice, which have 2- to 4-fold increased circulating PON-1 levels, showed significant decreases in atherosclerotic lesions compared to wild mice [11]. Further, PON-1-deficient mice demonstrate a higher risk for developing atherosclerosis compared to wild-type mice [34]. The HDL isolated from transgenic mice was also protected against LDL oxidation more effectively [11]. Given these potential atheroprotective effects, and the large burden of atherosclerotic CVD, a substantial effort has focused on elucidating the relevance of PON-1 in these settings. PON-1 may be a determinant of resistance to the development of atherosclerosis, perhaps by hydrolyzing phospholipid and cholesteryl-ester hydroperoxides. In line with these findings, previous studies have also highlighted the role of PON-1 in the settings of heart failure. Tang et al. reported that decreased serum PON-1 activity is a strong predictor of long-term adverse CVD outcomes [35]. Hammadah et al. examined the predictive significance of variations in PON-1 activity in heart failure patients [36]. After a follow-up of 2.8 years, decreased PON-1 activity was found to be linked with an increased risk of poor heart failure outcomes after adjustment for possible risk factors. Significant decreases in PON-1 activity throughout the follow-up were also linked with an increased risk of adverse heart failure outcomes [36]. These discoveries indicate that PON-1 activity might indeed serve as a prognostic biomarker in heart failure.

In the present study, the echocardiographic evaluation revealed that, in a well characterized salt-sensitive hypertensive renal disease model, the loss of PON-1 leads to compensated concentric LV hypertrophy as characterized by significant changes in end-diastolic and -systolic LV dimensions, significantly increased relative wall thickness, and significantly increased global systolic function. We used several direct measurements to evaluate systolic function including the percentage of fractional shortening, the mean velocity of circumferential fiber shortening, and the cardiac index, all of which clearly and consistently demonstrated differences in systolic function between the wild-type and PON-1 knock-out rats under salt-loaded conditions. Other investigators have shown a similar development of compensated concentric hypertrophy in congenic substrains of the Dahl salt-sensitive and Dahl salt-resistant rat strains, and these changes were associated with increased blood pressure [37]. We measured blood pressure via both tail-cuff plethysmography and radiotelemetry in a companion study, and demonstrated that that there is no difference in blood pressure between SS-PON-1 KO and SS-WT rats after 4 weeks on an 8% high-salt diet [21]. This suggests that the changes seen in the SS-PON-1 KO hearts are not the result of an increased afterload from elevated blood pressure, but rather from a more direct pathophysiologic effect of the loss of PON-1 on the cardiac structure and function. Interestingly, although almost all measures of cardiac structure and function were similar under baseline conditions, we did note that two important genes associated with calcium handling, SERCA2a and NCX-1, were decreased at baseline in the SS-PON-1 KO hearts. This suggests that the SS-PON-1 KO rats may have had some degree of subclinical cardiac dysfunction under basal conditions that was exacerbated by the high-salt diet-induced renal disease.

Many studies have also demonstrated the role of PON-1 in the pathogenic sequelae of CKD [38]. Patients with CKD demonstrate a significant decrease in PON-1 activity and concentration as assessed by PON-1 paraoxonase and arylesterase activities [20,39]. Reduced PON-1 activity in patients with CKD is associated with increased aortic stiffness and aortic strain, therefore increasing the risk of CVD [40]. Additionally, decreased serum PON-1 activity has been shown to predict an increased risk of long-term detrimental CVD outcomes (stroke, heart attack, or death) in CKD patients [7]. The serum PON-1 level and activity were found to be lower in patients with end-stage kidney disease [41,42], a finding that has been confirmed in multiple end-stage kidney disease cohorts [43,44,45]. These findings highlighted the importance of PON-1 in this high-risk population. However, whether PON-1 is mechanistically linked to adverse CVD events in CKD is not fully understood. Hence, in the current study, we examined the hypothesis that PON-1 is cardioprotective in the setting of CKD.

The direct examination of hearts revealed that, in addition to the functional and structural deficiencies noted above, SS-PON-1 KO rats developed myocardial damage as indicated by both the histologic and genetic assessment of cardiac fibrosis, inflammation, and hypertrophy. The histological evaluation of hearts collected from SS-PON-1 KO rats demonstrated increased inflammation and fibrosis. The heart tissues also showed significant increases in cardiac pro-inflammatory and pro-fibrotic gene expression, as well as a gene expression profile suggestive of heart failure and/or diastolic dysfunction [46,47]. Interestingly, the decreases in cardiac expression of both SERCA2a and NCX-1, as well as the echocardiographic and histologic phenotype, mirror the molecular, structural, and functional phenotype seen in other models of CKD, including 5/6 nephrectomy [48,49,50], which suggests that the high-salt diet-induced renal disease model may share a common phenotype with other CKD models of uremic cardiomyopathy. Taken together, the current study suggests that diminished levels of PON-1 promotes a pro-inflammatory and pro-fibrotic environment in the myocardium that leads to myocardial disease at the molecular, structural, and functional level. These findings extend our understanding of the role of PON-1 in CKD beyond its previously established anti-atherogenic role as an important accessory protein to the HDL, to now include anti-inflammatory and anti-fibrotic effects on the myocardium that have important implications for cardiac performance.

Our findings are similar to previous studies reporting that circulating PON activity is significantly decreased in patients with CKD [7] and heart failure [35]. These studies also revealed associations between lower-circulating PON activities and a higher cardiovascular mortality risk. Importantly, a recent study from our lab reported early mortality in SS-PON-1 KO rats on a high-salt diet (8%NaCl) without any mortality reported in SS-WT rats [21]. The study showed that SS-PON-1 KO rats had significant decreases in renal function along with increased renal inflammation and oxidative stress compared to SS-WT rats. SS-PON-1 KO rats also displayed evidence of increased renal injury represented by increased renal sclerosis, fibrosis, and acute tubular injury changes compared to SS-WT rats. Notably, SS-PON-1 KO and SS-WT rats demonstrated similar increases in blood pressure after high-salt feeding. Hence, similar to the cardiac phenotype, the detected differences in the renal phenotype between SS-PON-1 KO and SS-WT rats could not be linked to differences in blood pressure [21]. These results support experimental and clinical data that imply a protective role of PON-1 in CKD settings [7,8,51]. Marsillach et al. demonstrated that significant elevation in serum PON-1 activity could improve oxidative stress in pre-dialysis CKD patients [52]. These findings have been further confirmed by a recent study which showed that the decreased circulating lactonase activity of PON-1 predicts adverse clinical outcomes in patients with CKD [8].

At present, PON-1 can be considered as a major antioxidative enzyme in oxidative stress-related diseases given its unique properties. Studies related to dietary interventions meant to enhance PON-1 activity have reported significant antioxidant effects [53]. Pomegranate and some of its components demonstrated potent effects in enhancing PON-1 activity [54]. *Aronia melanocarpa* has also been reported to increase PON-1 expression and boost its antioxidative function [53,55]. Flavonoids are found in most of the related active extracts, with catechins and genistein being among the most likely elements for increasing PON-1 activity [55,56]. Prospective clinical trials of diets with these compounds merit study for their ability to modulate PON-1 activity and provide cardioprotection in at-risk patient populations such as those with CKD.

The current study highlights the roll of endogenous PON-1 in preventing cardiac disease in a well-established model of CKD. Our findings expand our current understanding of PON-1 as an anti-atherogenic enzyme, and suggest that PON-1 represents a novel, modifiable risk factor in CKD patients with important cardioprotective functions for cardiac structure and function. By augmenting anti-inflammatory and anti-fibrotic processes in the heart, PON-1 may serve as a potential therapeutic target to help reduce cardiac morbidity and mortality in this high-risk population. Future studies are needed to explore the mechanistic basis for the cardioprotective anti-inflammatory and anti-fibrotic roles of PON-1 in CKD.

## Figures and Tables

**Figure 1 biomedicines-10-02301-f001:**
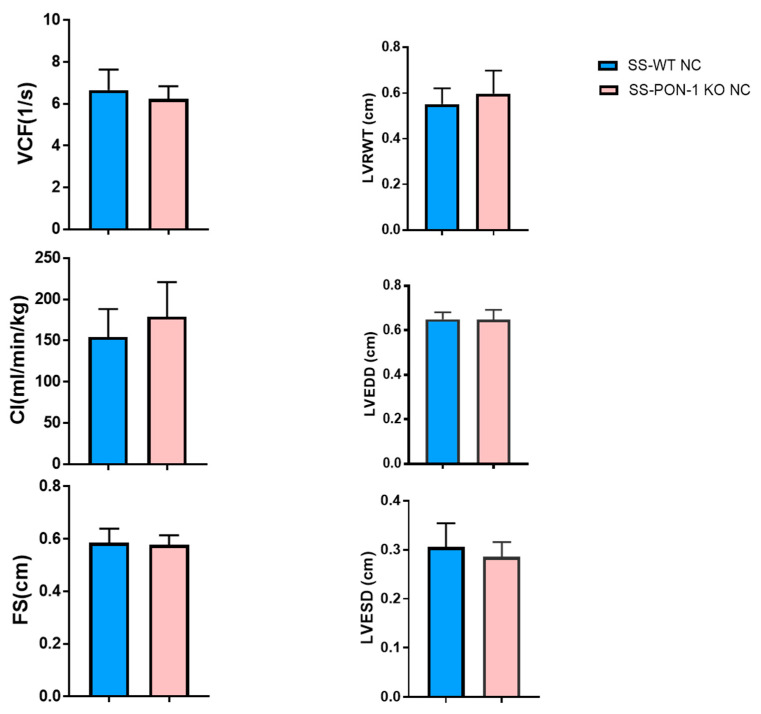
Echocardiographic evaluation at baseline (on normal chow) shows no differences in left ventricular geometry as well as in global systolic function (*N* = 8). Data are presented as the mean ± standard error of the mean. Student’s unpaired *t*-test was used to assess statistically significant differences between the groups. VCF: velocity of circumferential fiber shortening, CI: cardiac index, FS: fractional shortening, LVRWT: left ventricular relative wall thickness, LVEDD: left ventricular end-diastolic dimension, LVESD: left ventricular end-systolic dimension.

**Figure 2 biomedicines-10-02301-f002:**
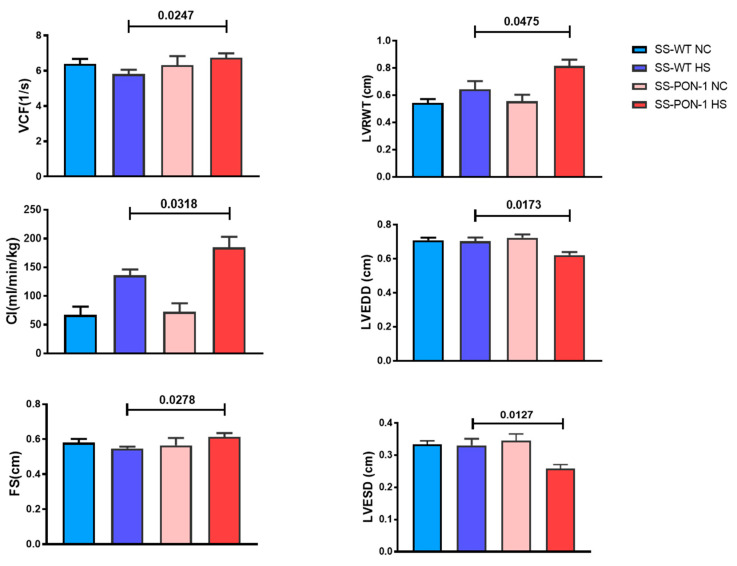
Echocardiographic evaluation after 4 weeks on 8% high-salt diet. SS-PON-KO HS rats demonstrated significantly decreased end-diastolic and -systolic dimensions, and a significantly increased relative wall thickness as well as increase in global systolic function as shown by increases in fractional shortening, velocity of fractional shortening, and cardiac index (*N* = 8). Data are presented as the mean ± standard error of the mean. One-way ANOVA and post hoc multiple comparison tests were used to assess statistically significant differences between the groups. VCF: velocity of circumferential fiber shortening, CI: cardiac index, FS: fractional shortening, LVRWT: left ventricular relative wall thickness, LVEDD: left ventricular end-diastolic dimension, LVESD: left ventricular end-systolic dimension.

**Figure 3 biomedicines-10-02301-f003:**
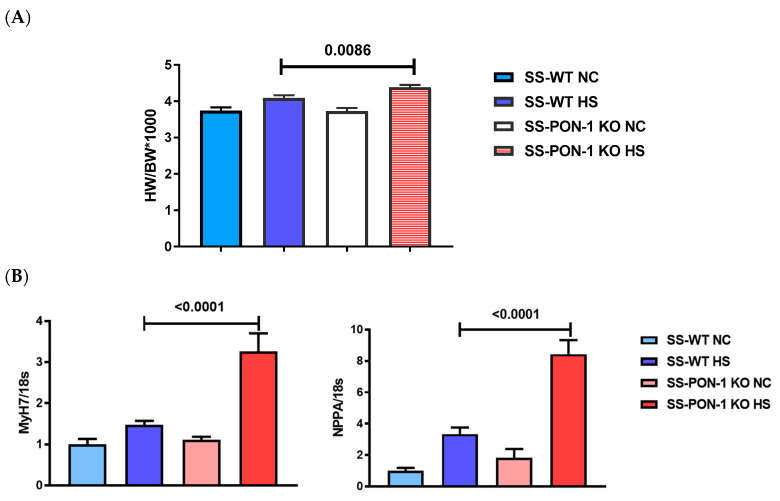
Targeted mutation of paraoxonase-1 leads to increased cardiac hypertrophy in a hypertensive renal disease model. (**A**) SS-PON-1 KO on high-salt diet demonstrate significant increase in heart-weight-to-body-weight ratio compared to SS-WT rats; (**B**) SS-PON-1 KO on high-salt diet demonstrate significant increase in cardiac hypertrophic gene expression compared to SS-WT rats (*N* = 8). Data are presented as the mean ± standard error of the mean. One-way ANOVA and post hoc multiple comparison tests were used to assess statistically significant differences between the groups.

**Figure 4 biomedicines-10-02301-f004:**
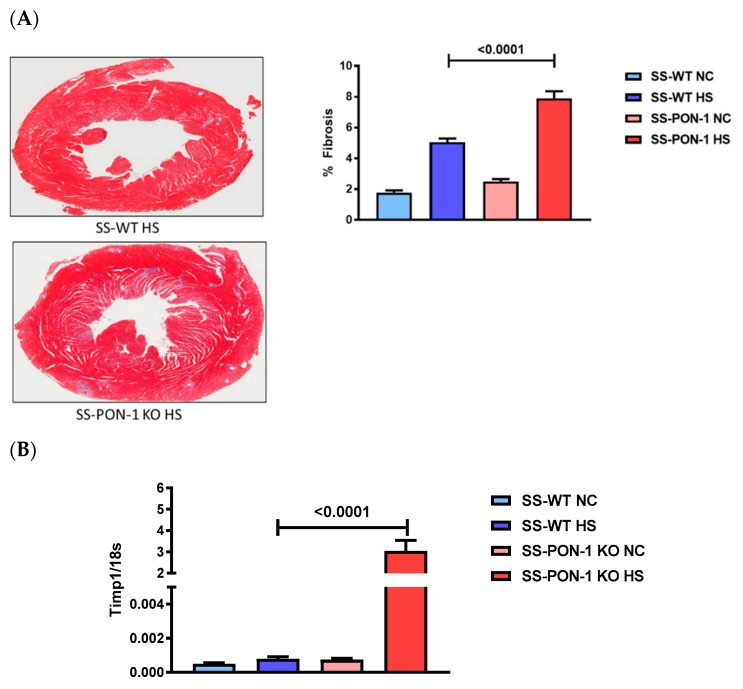
Targeted mutation of paraoxonase-1 leads to increased cardiac fibrosis in a hypertensive renal disease model. (**A**) Collagen area (pixels) (*N* = 8) and (**B**) gene expression of Timp-1 in cardiac tissue as assessed by RT-PCR from SS-PON-1 KO and SS-WT rats after 5 weeks of 8% high-salt diet (*N* = 8). SS-PON-1 KO HS rats display significant increase in cardiac fibrosis, compared to SS-WT HS rats following 5 weeks of 8% high-salt diet. Data are presented as the mean ± standard error of the mean. One-way ANOVA and post hoc multiple comparison tests were used to assess statistically significant differences between the groups.

**Figure 5 biomedicines-10-02301-f005:**
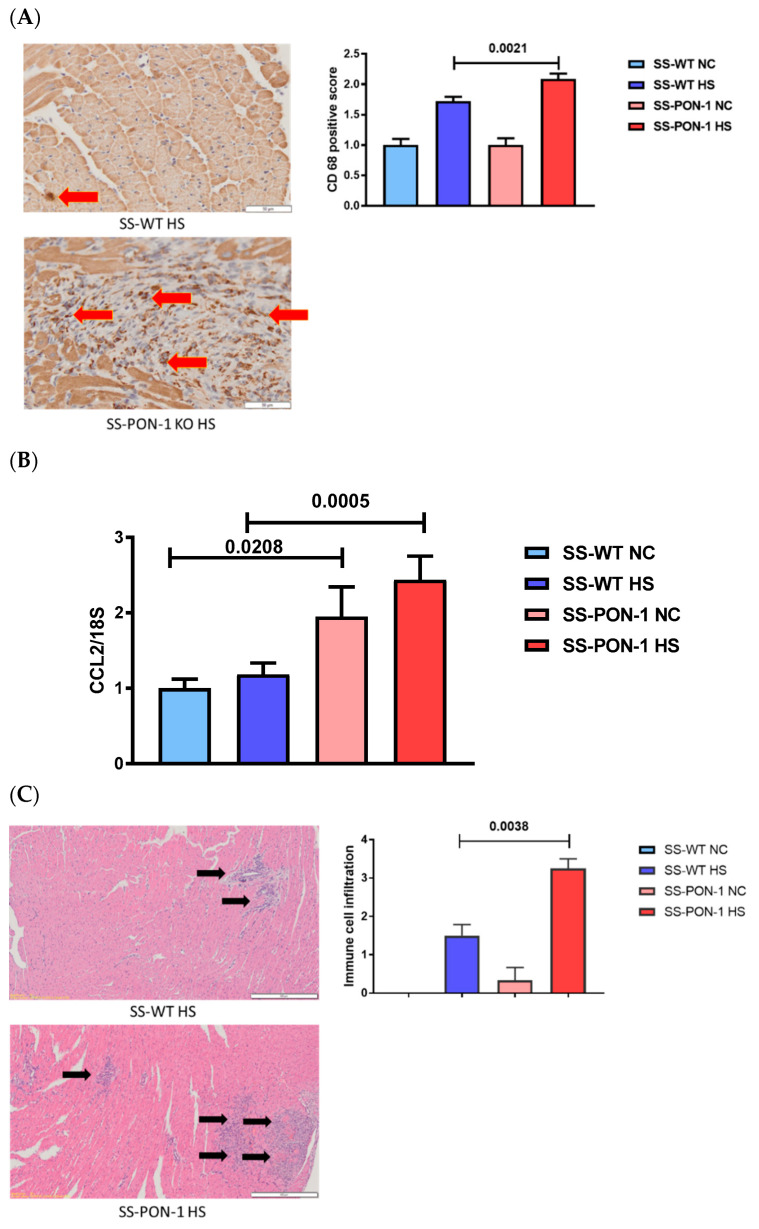
Targeted mutation of paraoxonase-1 leads to a significant increase in cardiac inflammation in a hypertensive renal disease model. (**A**) SS-PON-1 KO HS rats demonstrate significant increase in CD68-positive immune cells compared to SS-WT HS rats. (**B**) SS-PON-1 KO HS rats demonstrate significant elevation in inflammatory gene CCL2 expression as compared to SS-WT HS rats. (**C**) SS-PON-1 KO HS rats demonstrate significant increase in interstitial immune cell infiltration compared to SS-WT HS rats. Representative H&E histology (left) and quantification (right) (*N* = 8). Data are presented as the mean ± standard error of the mean. One-way ANOVA and post hoc multiple comparison tests were used to assess statistically significant differences between the groups.

**Figure 6 biomedicines-10-02301-f006:**
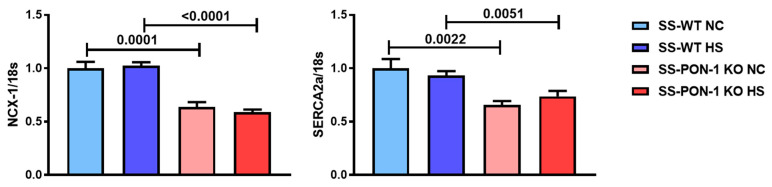
Targeted mutation of paraoxonase-1 contributes to decrease in left ventricular function gene expression in a hypertensive renal disease model. SS-PON-1 KO HS rats showed significant decrease in sarcoplasmic reticulum calcium ATPase 2A (atp2a2) and sodium–calcium exchanger (SLC8A1) gene expression compared to SS-WT rats following 5 weeks of 8% high-salt diet (*N* = 8). Data are presented as the mean ± standard error of the mean. One-way ANOVA and post hoc multiple comparison tests were used to assess statistically significant differences between the groups.

## Data Availability

The datasets generated and/or analyzed during the current study are available from the corresponding author on reasonable request.

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
