# Peer review of "Cardioprotective Role for Paraoxonase-1 in Chronic Kidney Disease"

_biomedicines, 2022, doi:10.3390/biomedicines10092301_

Round 1

Reviewer 1 Report

  1. Thank you for considering me to review the mansuscript entitling “Cardioprotective role for paraoxonase-1 in chronic kidney disease”. In this article, the authors investigated the possible cardioprotective role of PON1 in CKD, by using a Dahl Salt-Sensitive rat model of hypertensive renal disease. They concluded that PON1-KO led to a cardiac phenotype and compensated left ventricular hypertrophy with increased cardiac fibrosis and inflammation.

    I have the following comments:

    1. where have Dahl Rats for control groups been purchased or are they in-house bred as well to match?
    2. groups of the animal model should be better clarified in the mat&met section (abbreviations used in results do not match those in mat&met; eg you state NC and HS for normal chow and high salt, but do not define what later on will be referred to as SS rats, you should refer to it in text as you do in figures (even if at the point you are exclusively comparing SS and PON KO on NC))
    3. specify how echocardiographic measures were taken: long axis, short axis, lv tracing or mmode? reporting of methods is crucial
    4. to make a claim on actual systolic function EF has to be reported
    5. Did authors not assess diastolic function by E/A or E/e`?
    6. did authors not confirm hypertensive phenotype at all? tail cuff measures are to be reported
    7. stats: did authors check for normality of distribution?
    8. authors should consider adding panel letters and improve figure legends for all figures (really inconsistent)
    9. Authors should align figure axis, improve consistency of data representation (significance bar varies, font varies)
    10. Authors should consider reporting p values instead of stars
    11. Applied statistical tests should be reported in figure legends
    12. Did the authors report all statistically significant p values in figures? (eg CI in figure 2 and Fig 3, Fig 4A, Fig 5A)
    13. size bars are missing in representative images
    14. could authors confirm the decreased expression of NCX and Serca2a in SS PON1 KO rats upon HSD (and judging by the figures also NC)? Especially interesting in the setting of heart failure (where increase in NCX and decrease in Serca would be observed).
    15. Can authors perform any experiments to validate CKD phenotype?
    16. as it is known that upon prolonged HS feeding SS rats develop HF (both reports of HFpEF and HFrEF are published) it would have added value to both the study itself, PON1 as a prognostic biomarker and as a proof of principle to collect plasma at an earlier timepoint (eg the one studied here) and from there on in a 2 week interval and then keep rats on HS for another 6-8 weeks to determine the point when PON1 levels decrease.
    17. authors comment on not having done blood pressure measurements and refer to a previous study, which lacks blood pressure measurements of animals fed NC

    Overall, it appears that the paper lacks important in vivo data (echocardiography and blood pressure measurements) and has not been very carefully compiled ( assembly of figures and use of English language). In general, the report is purely descriptive and does not give any indication of the mechanism.

    In addition, it should be discussed to what extent the chosen feeding regime of the Dahl rats influences the development of their phenotype.

Author Response

We are grateful to the reviewers and editor for the thoughtful comments and insightful suggestions that helped us improve our manuscript considerably. As indicated in the responses below, we have taken all their valuable comments and suggestions into consideration in the revised manuscript. Please, note that the Reviewers’ comments are written in black, and our responses are written in
blue. 

Reviewer 2 Report

This manuscript talked about the PON-A as a potential antioxidant could have cardioprotective effect in chronic kidney disease through decreasing the cardiac fibrosis and inflammation. This work is interesting and meaningful, but there are many points should be further considered.

1. The authors just showed the mRNA expression level of key genes, such as Figure 3B, Figure 4B and Figure 6, which may not fully reflect the relevance to phenomena.

2. The authors considered the influence of SS-PON-1 to cardiac fibrosis and cardiac inflammation. The severity of cardiac fibrosis is related with cardiac fibrosis, how they consider about this relationship?

3. This manuscript did many experiments to show the changes in SS-PON-1 KO rats, but its really hard for me to get the correlation among measures.

Author Response

(The authors gave the same response as above.)

Round 2

Reviewer 1 Report

I'd like the thank the authors for addressing the issues raised in my first revision. 

For some figures (e.g. Figure 3A, Figure 5B) the font is different from the other graphs. I suggest to change this. 

Author Response

For some figures (e.g. Figure 3A, Figure 5B) the font is different from the other graphs. I suggest to change this. 

Response: We thank reviewer for this comment, we have now modified the figure in the manuscript.

Reviewer 2 Report

This revised manuscript can be now considered acceptable for publication.

Author Response

This revised manuscript can be now considered acceptable for publication.

Response: We thank the reviewer for the positive comment.